# Learning Geometry Consistent Neural Radiance Fields from Sparse and Unposed Views

## ABSTRACT

The recent progress in novel view synthesis is attributed to the Neural Radiance Field (NeRF), which requires plenty of images with precise camera poses. However, collecting available dense input images with accurate camera poses is a formidable challenge in real-world scenarios. In this paper, we propose Learning Geometry Consistent Neural Radiance Field (GC-NeRF), to tackle this challenge by jointly optimizing a NeRF and camera poses under sparse (as low as 2) and unposed images. First, GC-NeRF establishes the geometric consistencies in the image-level, which produce photometric constraints from inter- and intra-views for updating NeRF and camera poses in a fine-grained manner. Second, we adopt geometry projection with camera extrinsic parameters to present the region-level consistency supervisions, which construct pseudo-pixel labels for capturing critical matching correlations. Moreover, GC-NeRF presents an adaptive high-frequency mapping function to augment the geometry and texture information of the 3D scene. We evaluate the effectiveness of GC-NeRF, which sets a new state-of-the-art in the sparse view jointly optimized regime on multiple challenge real-world datasets.

## CCS CONCEPTS

• **Computing methodologies → Image-based rendering**;

## KEYWORDS

Neural Radiance Fields, Volume Rendering, Sparse and Unposed Views, Geometric Consistency

**ACM Reference Format:**
Anonymous Author(s). 2024. Learning Geometry Consistent Neural Radiance Fields from Sparse and Unposed Views. In *Proceedings of Proceedings of the 32th ACM International Conference on Multimedia (ACM MM '24)*. ACM, New York, NY, USA, 10 pages. https://doi.org/XXXXXXX.XXXXXXX

## 1 INTRODUCTION

Novel view synthesis in real-world scenes is a crucial task in computer vision, requiring accurate 3D geometric reconstruction and realistic modeling of appearance textures. The recent advancements in Neural Radiance Field (NeRF), leveraging its powerful implicit scene representation capabilities, have showcased its immense potential in the field of novel view synthesis [21].

Nevertheless, the remarkable performance of NeRF is heavily contingent on two critical requirements: dense input views and precise camera poses. These requirements impose significant limitations on the practical applicability of NeRF in real-world scenarios. In many cases, such as VR/XR applications, the availability of input images becomes sparse, often leaving only a few RGB images captured from specific viewpoints [39]. Consequently, NeRF may suffer from overfitting and produce unsatisfactory or even erroneous synthesized images [14, 23]. Additionally, although mature and convenient community repositories like COLMAP [30] exist to assist with camera pose estimation, which is an essential preprocessing step before NeRF training, the non-differentiable nature of these methods presents challenges when integrating them with NeRF. This limitation hinders the seamless integration between pose estimation and NeRF, impeding the joint optimization of NeRF and camera poses for improved performance. Furthermore, it increases NeRF's reliance on third-party libraries, consequently hindering research progress and practical deployment, particularly in scenarios with sparse views where accurate camera pose estimation using COLMAP becomes challenging [50].

Numerous researchers have made efforts to reduce NeRF's dependence on accurate camera poses. These efforts can be broadly classified into two approaches: deep network-updated pose optimization and estimator-based pose prediction. The former treats camera poses as learnable parameters, formulating novel view synthesis as a joint optimization problem of NeRF and camera poses [1, 13, 17, 42]. The latter approach involves incorporating a separate pose estimator to predict camera poses corresponding to the input images before feeding them into NeRF [3, 10, 29, 48]. While both approaches have shown promising potential, they heavily rely on having a sufficient number of input views, and their performance significantly deteriorates when dealing with sparse view images. Because the under-constrained nature of the 3D space limits the convergence of the deep network module to an optimal state due to insufficient training data coverage.

Recently, a growing number of researches dedicated to enhancing the performance of NeRF under sparse view settings. Some methods focus on training conditional neural field models on large-scale datasets [2, 47]. Additionally, various regularization techniques have been proposed to address the challenges posed by sparse views in terms of the color and geometry for the scene [6, 14, 23]. Although these researches show impressive performances, they have not completely gotten rid of the precise camera poses. It is worth noting the closest to our work is SPARF [39], which is also under sparse view conditions. However, the latter excessively relies on the initialization of camera poses, training NeRF under noisy pose settings. It does not fundamentally alleviate the dependence on camera poses to achieve joint optimization of NeRF and poses.

In this paper, we propose the *Learning Geometry Consistent Neural Radiance Field* (GC-NeRF), to optimize NeRF and camera poses

                    

jointly from sparse input views. We illustrate the framework of the GC-NeRF in Figure 1, where the whole pipeline consists of the *Image Level Consistency*, *Region Level Consistency*, and *Adaptive High Frequency Positional Encoding*. For the *Image Level Consistency*, we establish correlations from intra- and inter-view images for constructing different color labels. Specifically, we first supervise the quality of image synthesis with the standard photometric rendering. Till here, we establish the projection geometric relationship between the source and target viewpoints, and provide pseudo-labels from the source (target) view to the rendered image of the target (source) view. This approach constructs an adjacent photometric supervision to enhance the learning capability of camera poses in the joint optimization problem. For the *Region Level Consistency*, GC-NeRF localizes key regions of the source and target views by the correspondences between matching points across image pairs. Then we re-project the matching points to capture the pixel position labels and 3D coordinates labels. Consequently, GC-NeRF establishes a novel photometric rendering loss, matching projected supervision, between the projected pixels and the labeled pixels in the rendered images. This effectively augments the joint optimization performance. For the *Adaptive High Frequency Positional Encoding*, the proposed adaptive fusion random Fourier features (AdaRFF) strategy can map the coordinates to a high-dimensional space. It will adaptively filter high-frequency features into MLP network so that benefiting the NeRF model to render high-quality novel view images even with sparse input views.

We evaluate GC-NeRF and the state-of-the-art baselines, which effectively outperform various methods to achieve the SOTA performance on the Tanks and Temples [15], LLFF [33], and NeRF Real 360 [21] datasets for sparse and unposed images NeRF optimization. The main contributions are summarized as follows:

- We advocate the idea of GC-NeRF to optimize NeRF model from sparse and unposed images.
- We propose effective supervisions of GC-NeRF, the image- and region-level consistencies, for establishing geometric correspondences for input views and augmenting the joint optimization capabilities of NeRF model and camera poses.
- We present a novel position encoding method, adaptive fusion random Fourier features, for passing input coordinates to a higher dimension space and performing satisfactorily at representing high-frequency variation in the scene.
- We conduct comprehensive analyses on GC-NeRF, which achieves the state-of-the-art results in several real-world scenarios, demonstrating the effectiveness of our method.

## 2 RELATED WORK

We mainly introduce two different aspects of researches for NeRF, i.e., the novel view images synthesis and the camera and extrinsic parameters estimation, which are relevant to our method.

**Novel View Synthesis for NeRF.** In recent years, Neural Radiance Field (NeRF) [21] has become the favored image rendering because of its remarkable implicit scene representation. Numerous methods [2, 31, 32, 40, 45, 47] are presented to improve the performance of NeRF. Typically, a number of researchers have integrated depth or point cloud to supervise the training and rendering

stage [7, 26]. Moreover, an additional regularisation is another effective strategy to augment the photometric quality [14, 43, 49]. Recently, many innovative constructs have accelerated training and rendering [9, 11, 22]. All of these models achieve fantastic results. However, most of them rely on sufficient images, which are indispensable to pre-computed camera extrinsic parameters.

Different from the above approaches, we exploit the geometric consistency relations between input views, which enables the novel view synthesis for implicit scene representation with only two unposed images.

**Pose Estimation for NeRF.** Weaning NeRF off camera parameter preprocessing is a significant attempt for optimizing NeRF nowadays [20, 44, 46]. Based on the SLAM technique, some researchers employ RGB-D data or exploit SLAM's tracking capabilities to pre-calculate the camera poses [27, 36, 53]. Moreover, the other methods focus on learning NeRF model and camera poses jointly [1, 42]. iNeRF [18] sets camera poses as learnable parameters, which can be optimized by minimizing the residual between the predicted and ground truth pixels. BARF [17] combines bundle-adjustment strategy and NeRF to solve the joint problem of pose estimation and NeRF model. SCNeRF [13] jointly learns the scene and the camera parameters by self-calibration algorithms. GARF [4] employs Gaussian activation to improve the joint optimization of pose and scene. A mount of works [3, 10, 29, 48] adopt the generative network or pose/depth estimators to predict camera poses before learning the camera and NeRF model.

Although these solutions show infusive performance, they still suffer from indispensable pose initialization or sufficient images. Toward the end, We jointly optimize camera parameters and NeRF by image- and region-level supervision in sparse views settings.

## 3 PRELIMINARIES

Neural radiance field (NeRF) [21] represents continuous scenes as regressing from a single 5D coordinate to view-dependent RGB color, which can be optimized by minimizing photometric supervision as follows:

$$\hat{\Theta} = \underset{\Theta}{\arg\min} \mathcal{L}_{rgb}(\hat{I} \mid I, [\mathbf{R}|\mathbf{t}], \mathbf{K}), \tag{1}$$

where $\Theta$ is a basic MLP network. $\hat{I}$ is the predicted image. $[\mathbf{R}|\mathbf{t}] = \Gamma \in SE(3)$, $\mathbf{R} \in SO(3)$, $\mathbf{t} \in \mathbb{R}^3$, and $\mathbf{K} \in \mathbb{R}^{3\times3}$ represented camera extrinsic and intrinsic parameters, respectively. Particularly, we can synthesize the color $C$ of pixel $\mathbf{p}$ in $\hat{I}$ along the ray from $t_n$ to $t_f$ as follows:

$$C(\mathbf{r_p}) = \int_{t_n}^{t_f} T(t)\sigma(\mathbf{r_p}(t))\mathbf{c}(\mathbf{r_p}(t), \mathbf{d_p})dt, \tag{2}$$

where $\sigma(\mathbf{r_p}(t)) \in \mathbb{R}$ and $\mathbf{c}(\mathbf{r_p}(t), \mathbf{d_p}) \in \mathbb{R}^3$ denote volume density and RGB color of 3D location $\mathbf{x} = \mathbf{r_p}(t)$, respectively. $\mathbf{d_p} \in \mathbb{S}^2$ is the ray direction, which is calculated by camera parameters $[\mathbf{R}|\mathbf{t}]$ and $\mathbf{K}$, and pixel $\mathbf{p}$. $T(t) = \exp(-\int_{t_n}^{t} \sigma(\mathbf{r_p}(s))ds)$ indicates the accumulated transmittance. Notably, we can predict a view-invariant volume density $\sigma$ and a view-dependent color $\mathbf{c}$ according to 3D location $\mathbf{x}$ and view direction $\mathbf{d}$ by an MLP neural network $f_{\Theta}$ such that:

$$[\sigma, \mathbf{c}] = f_{\Theta}(\gamma(\mathbf{x}), \gamma(\mathbf{d})), \tag{3}$$

where $\gamma(\cdot) : \mathbb{R}^3 \rightarrow \mathbb{R}^{2L}$ denotes the positional encoding function under $L$ harmonics. Finally, a photometric rendering loss could optimize $\Theta$ by Eq. (1).

## 4 METHOD

In this section, we first elucidate the motivation and method overview in Section 4.1. Then we report the framework of GC-NeRF in Section 4.2 – 4.6.

### 4.1 Motivation and Overview

This work aims to improve novel view synthesis by NeRF from sparse and unposed camera images. Specifically, we address on the challenging task of synthesizing novel views using only sparse input images (as low as 2 views), without requiring accurate or specific initial camera poses.

This results in two critical challenges. Firstly, only a few input images for NeRF [21] in training will overfit instantly, which leads to terrible rendering quality of meaningful 3D geometry, even with ground-truth camera parameters [39]. Secondly, current jointly optimizing for pose and NeRF approaches focus on dense views of 3D scenes [1], along with relying on proper pose initialization. Especially, the above challenges will restrain each other, such as the existing photometric supervision is weak to perceive 3D geometric scenes from sparse views. Thus failure to get rid of poses under sparse input views.

Towards this end, we propose a novel Learning Geometry Consistent Neural Radiance Field method, dubbed GC-NeRF, which combines image- and region-level consistencies to overcome the joint optimization problem. And an adaptive positional encoding strategy is designed for augmenting high-frequency perception.

Specifically, given a pair of images $I_S$ and $I_T$, along with the correspondences between matching points in two views, we establish geometric consistency at both the image- and region-level.

For the image-level consistencies, we propose an adjacent photometric supervision that leverages photometric rendering in standard NeRF. Specifically, we employ epipolar geometry to provide pseudo color labels $\mathbf{p}_{S \rightarrow T}$ and $\mathbf{p}_{T \rightarrow S}$ for the rendered images $\hat{I}_T$ and $\hat{I}_S$, respectively, from the corresponding target and source views. The adjacent photometric supervision enables to optimize NeRF model and camera poses simultaneously.

For the region-level consistencies, we consider the matching points relations of the input image pair $I_S$ and $I_T$ as re-projection labels for key pixels. As the Lambertian model, we aim to ensure that the re-projection of a pixel $\mathbf{p}_S$ onto the target image $\tilde{\mathbf{p}}_{S \rightarrow T}$ ideally corresponds to the matched pixel with the same color. We establish this geometric relationship on the rendered images $\hat{I}_S$ and $\hat{I}_T$ of the image pair, facilitating the joint optimization of the NeRF model and camera poses.

Finally, we introduce the adaptive fusion random Fourier features (AdaRFF) positional encoding module based on the random Fourier features (RFF) transformation, which adaptively augments high-frequency variations of the scene when mapping coordinates to a higher-dimensional space. This framework overcomes the limitations of previous sparse view setting approaches, improving not only the global geometry but also the high-frequency details and appearance of the rendered images.

In the following, we elaborate the pipeline of GC-NeRF, including the *Image Level Consistency, Region Level Consistency, Adaptive High Frequency Positional Encoding.*

### 4.2 Poses Initialization

To solve the joint optimization problem of NeRF model and camera poses, we first initialize camera pose parameters $\mathbf{R} \in SO(3)$ as the quaternions $\mathbf{q}(w, x, y, z)$ to represent a 3D rotation. Different from the previous works to initialize with the axis-angle [42], the 6-vector representations [13, 39] or the Euler angles [1], the quaternions strategy only has 4 parameters rather than 6 without Gimbal lock problem [34], which alleviates the gradients backward to update the three degrees of freedom rotation effectively. The transformation from a quaternion $\mathbf{q}(w, x, y, z)$ to a rotation matrix $\mathbf{R} \in \mathbb{R}^{3 \times 3}$ is as follows:

$$\mathbf{R} = \begin{bmatrix} 1 - 2(y^2 + z^2) & 2(xy - zw) & 2(xz + yw) \\ 2(xy + zw) & 1 - 2(x^2 + z^2) & 2(yz - xw) \\ 2(xz - yw) & 2(yz + xw) & 1 - 2(x^2 + y^2) \end{bmatrix}. \quad (4)$$

### 4.3 Image Level Consistency

The image-level consistency consists of two aspects: the intra-view relations between rendering images and ground truth (GT) images at the same poses, as well as the inter-view consistency associations between rendered images and the GT from adjacent images.

**Image Photometric Supervision.** The image photometric loss is a critically acclaimed strategy to bridge the gap between the rendered and the GT images at the same viewpoint, which adopts the squared error of intra-view pixel color to supervise NeRF model convergence [1]. The details are as follows:

$$\mathcal{L}_{ips} = \sum_{\mathbf{p} \in \mathcal{R}} \|\hat{C}(\mathbf{p}) - C(\mathbf{p})\|_2^2, \quad (5)$$

where $\mathcal{R}$ is the set of rays in each batch. $C(\mathbf{p})$ and $\hat{C}(\mathbf{p})$ denote the ground truth and predicted RGB colors of pixel $\mathbf{p}$ for ray $\mathbf{r}$ by Eq. (2), respectively.

**Adjacent Photometric Supervision.** Directly employing sparse or unposed image settings will lead to corrupted NeRF [21] collapsing with overfitting. Because only the image photometric loss (Eq. (5)) is impossible to support the harsh demands with the limited ground-truth from the intra-view pixel color. We present a training supervision upon inter-view images, adjacent photometric loss, to borrow enough constrains from the adjacent view for augment perceiving multi-view consistency for 3D scenes.

We get the inspiration from epipolar geometry to augment the geometric correspondences of input views. Given two images $I_S$ and $I_T$ and their rendered images $\hat{I}_S$ and $\hat{I}_T$ with source and target viewpoints $\Gamma_S$ and $\Gamma_T$, respectively. We can warp $\hat{I}_S$ to pose $[\mathbf{R}_T | \mathbf{t}_T]$ by the transformation as follows:

$$\begin{aligned} \overline{Q}_S &= \overline{\begin{bmatrix} \mathbf{R}_S & \mathbf{t}_S \end{bmatrix}}^{-1}(D_S \mathbf{K}_S^{-1} \overline{\mathbf{p}}_S), \\ \overline{Q}_{S \rightarrow T} &= \overline{\begin{bmatrix} \mathbf{R}_T & \mathbf{t}_T \end{bmatrix}} \overline{Q}_S, \\ \overline{\mathbf{p}}_{S \rightarrow T} &= \text{Norm}(\mathbf{K}_T Q_{S \rightarrow T}), \end{aligned} \quad (6)$$

where $\mathbf{p}_S \in \mathbb{R}^{2 \times 1}$ is the pixel of image $I_S$. $\text{Norm}(\cdot)$ denotes the scale normalization. $\overline{(\cdot)}$ depicts the homogeneous representation. $D_S$ is the depth of $\mathbf{p}_S$. $\mathbf{p}_{S \rightarrow T}$ denotes the warped pixel in target

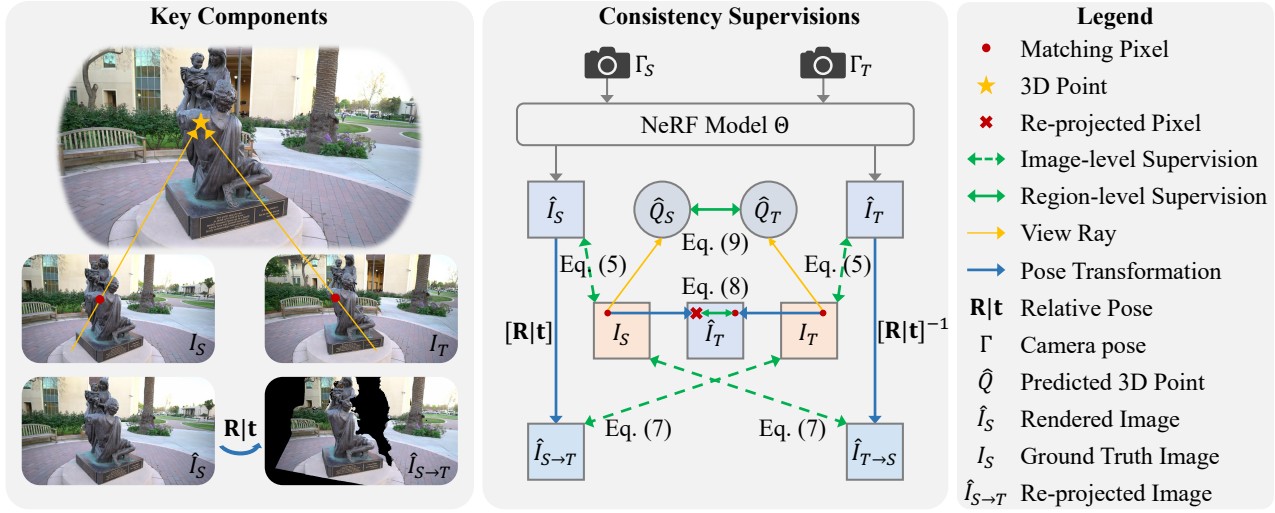

**Figure 1: Our approach GC-NeRF for joint camera pose and NeRF model training from only sparse and unposed input views. We first establish geometric relations between the image pair to construct photometric supervisions from intra- and inter-view in the image-level (Eq. (7)). For region-level consistencies, we present a novel photometric rendering loss (matching projected loss) between the projected pixels and the matching pixels in the rendered images (Eq. (8)). We also bridge the gap of 3D points for matching pixels to supervise the optimization of camera extrinsic parameters and the density field (Eq. (9)).**

viewpoint $\Gamma_T$. After that, we can obtain the warped image $\hat{I}_{S \to T}$. To optimize pose and NeRF model jointly, we then assess the squared error between $\hat{C}(\mathbf{p}_{S \to T})$ in $\hat{I}_{S \to T}$ and the pseudo color label $C(\mathbf{p}_T)$ in $I_T$ as follows:

$$\mathcal{L}_{aps} = m \odot \sum_{\mathbf{p} \in \mathcal{R}} \|\hat{C}(\mathbf{p}_{S \to T}) - C(\mathbf{p}_T)\|_2^2, \qquad (7)$$

where $m$ is the mask to filtrate the projected pixels out-of-bounds [5]. We also calculate the errors between $\hat{C}(\mathbf{p}_{T \to S})$ in $\hat{I}_{T \to S}$ and the pseudo color label $C(\mathbf{p}_S)$ in $I_S$. The Eq. (6) is differentiable during the training stage. Thus the adjacent photometric supervision can pass the gradients to camera parameters $[\mathbf{R}|\mathbf{t}]$ of image pairs, while refining rendering quality by the backward for NeRF model.

Of utmost importance, the occlusion problem of multi-view images will transform distorted or wrong semantics images $\hat{I}_{S \to T}$ and $\hat{I}_{T \to S}$, which affect the accuracy of pseudo color labels sincerely. We present a technique to compare the projected depth $d_r$ from $\mathbf{p}_S$ to $\mathbf{p}_{S \to T}$ and the rendered depth $d_p$ of $\mathbf{p}_{S \to T}$ in $I_T$. Then we reserve the pixels with a threshold $\beta < \frac{d_r}{d_p} < \frac{1}{\beta}$ ($\beta \le 1$). Till here, the unambiguous gradients that are calculated from the adjacent view by Eq. (7) can further facilitate the joint optimization task.

### 4.4 Region Level Consistency

The image-level consistency favors the global supervision for 3D scenario perception across input views. However, we assume that access to only sparse views (i.e., 2 views) with unposed camera pose predicts (i.e., initialized with identity). Thus the above consistency losses are not sufficient to fulfill the joint optimization completely. We present additional supervisions from the other aspect, region-level consistencies, to augment 3D geometric analysis robustly and consistently.

**Matching Association.** The region-level consistency depicts the geometric affiliation of key pixels among multi-view images. We adopt the key points matching strategy, a pre-trained correspondence regression method (LoFTR [37]), to extract a matching point $\mathbf{p}_T$ for each pixel $\mathbf{p}_S$. Empirically, any learned based [8, 12] or classical [19, 28] matching solver can be qualified to establish matching associations of input views. Logically, the greater the number of matching points with high accuracy, the greater the facilitation. However, for augmenting NeRF joint optimization significantly, we only need 1% - 2% of the total pixel count as matching points to be converged, rather than 10% or even more matching points, which will be illustrated in Section 5.4.

**Matching Projected Supervision.** The core idea is to calculate whether the color of the re-projected pixel $\tilde{\mathbf{p}}_S$ is consistent with the color of the matching point $\mathbf{p}_T$ in $\hat{I}_T$. We sample an image pair $I_S$ and $I_T$. Then we can obtain the matching sequence $\mathcal{M}$. Specifically, for pixel $\mathbf{p}_S$ in the viewpoint $\Gamma_S$, we will get the corresponding projected pixel $\tilde{\mathbf{p}}_S$ following Eq. (6) in the viewpoint $\Gamma_T$. In the rendering image $\hat{I}_T$, the matching pixels $\mathbf{p}_T$ and $\tilde{\mathbf{p}}_S$ manifest the same color while camera pose parameters and NeRF model are convergence. Therefore, we narrow the pose discrepancy by supervising the matching pixel colors' consistency. We formulate our matching projected loss as follows:

$$\mathcal{L}_{mps} = \sum_{\mathbf{p} \in \mathcal{M}} c_{\mathbf{p}} \|\hat{C}(\tilde{\mathbf{p}}_S) - \hat{C}(\mathbf{p}_T)\|_2^2, \qquad (8)$$

where $\tilde{\mathbf{p}}_S$ is the re-projected pixel of $\mathbf{p}_S$ in viewpoint $\Gamma_T$. $\mathbf{p}_T$ denotes the matching pixel of $\mathbf{p}_S$ in viewpoint $\Gamma_T$, and $c$ is the confidence of the matching points.

For the multi-view occlusion problem, previous work [5] could alleviate it properly by the mask strategy. However, under the

sparse and unposed images settings, the mentioned scheme can not distinguish the occlusion phenomenon such two aspects: firstly, it is a real occlusion circumstance in the real world scenario. Secondly, it is most likely that the pose is inaccurate because the NeRF model and camera parameters have not converged, thereby calculating "pseudo-occlusion". This greatly hinders NeRF's ability to perceive scene 3D geometric information. Specifically, GC-NeRF establishes matching projected consistency supervision by matching points can effectively alleviate this problem.

Moreover, compared with the re-projected error [39], the matching projected supervision has ability to update the fully connected layers for the density $\sigma$ and color $\mathbf{c}$ of 3D locations, and camera parameters $\Gamma$ simultaneously. However, the former only learned the density and camera parameters, which enjoys a weaken capacity for NeRF and pose joint optimization. We therefore present this matching projected supervision for region consistency.

**Space Similarity Supervision.** Different from the matching projected supervision in Eq. (8), we also build the consistency loss function directly based on projected 3D coordinates to enhance NeRF for perceiving the region's geometric information. Benefiting from the matching sequence $\mathcal{M}$, the 3D coordinates $\hat{Q}_S$ and $\hat{Q}_T$ can be predicted from the corresponding matching points under $\Gamma_S$ and $\Gamma_T$. Before the model converges, there will be a spatial position deviation in the 3D coordinates calculated for the same pair of matching points $\mathbf{p}_S$ and $\mathbf{q}_T$ from two viewpoints. We present the 3D space similarity supervision to reduce this bias. The specific calculation is as follows:

$$\mathcal{L}_{sss} = \frac{1}{|\mathcal{M}|} \sum_{\mathbf{p} \in \mathcal{M}} c_{\mathbf{p}} \|\hat{Q}_S - \hat{Q}_T\|_2^2, \tag{9}$$

where $\hat{Q} \in \mathbb{R}^{3 \times 1}$ is calculated by Eq. (6), which omits subscript index. Due to the space similarity supervision, We can make the two point clouds infinitely close until they are almost identical, that is the camera pose is fitted as accurately as possible.

## 4.5 Adaptive High Frequency Positional Encoding

NeRF heavily relies on a positional encoding module to map the inputs into a high-dimensional space. However, recent work by Rahaman [24] has revealed that deep networks exhibit a bias toward learning lower-frequency functions. Thus a simple Fourier feature transformation that is commonly used in the standard NeRF [21], which fails to capture high-frequency information effectively [38].

The failure of the positional encoding module drastically reduces the ability to represent the details of geometry and texture of the scene (e.g. the rendered appearance becomes over-smoothed). This limitation is particularly severe for GC-NeRF, because it stems from the photometric and geometric constraints of the quality of rendered pixels' color. This phenomenon hinders the effectiveness of the image- and region-level geometric consistency supervisions.

Drawing inspiration from [25], we propose a novel approach called adaptive fusion random Fourier features (AdaRFF), by combining RFF with the conventional Fourier feature (FF) mapping. The presented adaptive fusion RFF component aims to enhance the capacity to learn high-frequency geometry and texture, thus enhancing the quality of scene perception and rendering. The RFF

mapping function is calculated as follows:

$$\beta(\mathbf{x}) = [sin(2^0 \pi \mathbf{l}_0 \mathbf{x}), \ \cos(2^0 \pi \mathbf{l}_0 \mathbf{x}), \ \cdots,$$
$$sin(2^{L-1} \pi \mathbf{l}_{L-1} \mathbf{x}), \ \cos(2^{L-1} \pi \mathbf{l}_{L-1} \mathbf{x})], \tag{10}$$

where $\mathbf{l} \in \mathbb{R}^3$ denotes the probability space, which is equipped with the Gaussian measure and non-zero standard deviation. We follow [21] to set $L = 10$. Then we combine the conventional Fourier feature with RFF to introduce adaptive fusion weight coefficients $\mathbf{g}$, which can dynamically filtrate the desired frequency bands for optimal scene representation. The details are as follows:

$$\mathbf{g} = \sigma(\mathbf{W}_1 \beta(\mathbf{x}) + \mathbf{W}_2 \gamma(\mathbf{x})), \tag{11}$$

$$\psi(\mathbf{x}) = \mathbf{g} \cdot \beta(\mathbf{x}) + (1 - \mathbf{g}) \cdot \gamma(\mathbf{x}), \tag{12}$$

where $\mathbf{g} \in \mathbb{R}^{2L}$ denotes the adaptive coefficients. $\sigma(\cdot)$ is the sigma function. $\mathbf{W}_1, \mathbf{W}_2 \in \mathbb{R}^{2L \times 2L}$ depict learnable parameters. $\cdot$ is the Hadamard product. And $\beta(\cdot)$ and $\gamma(\cdot)$ denote RFF and the conventional Fourier feature mapping functions. Thus, GC-NeRF can choose the proper frequency bands and scale adaptively to reconstruct a high-fidelity neural scene in novel view synthesis.

## 4.6 Training Framework

**Objective Function.** Assembling all loss terms, our overall training objective loss function is formulated as follows:

$$\mathcal{L} = \mathcal{L}_{ips} + \lambda_1 \mathcal{L}_{aps} + \lambda_2 \mathcal{L}_{mps} + \lambda_3 \mathcal{L}_{sss} \tag{13}$$

where $\lambda_1$, $\lambda_2$, and $\lambda_3$ are predefined weighting coefficients for each loss terms. By minimising the Eq. (13), our approach returns optimised camera poses $\Gamma$ and NeRF's MLP network $\Theta$.

**Training Pipeline.** For GC-NeRF, there is only one MLP network rather than a coarse and a fine networks. The training is split into two stages. First, we only use the image photometric supervision Eq. (5) to assist camera pose parameters representing discrepancies between each other. Then Eq. (5) - Eq. (9) supervise together to establish multi-view geometry consistency for optimizing pose and NeRF parameters jointly. Second, we freeze camera pose parameters and finetune the NeRF model with four supervisions until rendering high quality novel view images. Simultaneously analyzing image pairs does not result in excessive computation for the training pipeline, as there are shared operations in color and depth rendering.

## 5 EXPERIMENTAL RESULTS

### 5.1 Experimental Settings

**Datasets.** We evaluate all baselines and our presented GC-NeRF on the **Tanks and Temples** [15], **LLFF** [33], and **NeRF real 360** [21] real-world scene datasets, where we report the results of unknown and known pose methods, respectively. **Tanks and Temples** includes both outdoor scenes and indoor environments. Following [1], we use 8 scenes to evaluate all methods. We chose scenes captured at both indoor and outdoor scenarios, and 1/8 of the images in each sequence are used for novel view synthesis. **LLFF** contains 8 sets of different real-life scenes, which are primarily captured in the forward-facing direction. Consistent with [21], we evenly select the test set by extracting every 8th image from the video list. **NeRF Real 360** is composed of 2 different complex object-level scenes, which are captured by a set of inward-facing views. Each scene has

been taken from 99 and 116 positions, corresponding to the number of RGB images. We follow the protocol of [21] and evaluate 1/8 of the views for novel view synthesis.

**Metrics.** We report the performances of Novel View Synthesis, Camera Pose Estimation, and Depth Estimation in three benchmarks. For Novel View Synthesis, following [1, 13, 17], we select Peak Signal-to-Noise Ratio (PSNR), Structural Similarity Index Measure (SSIM) [41], and Learned Perceptual Image Patch Similarity (LPIPS) [51] standard evaluation metrics. For Camera Pose Estimation, we employ Relative Pose Error (RPE) to calculate the relative pose errors between image pairs, which includes rotation and translation errors [16, 35, 52]. Notably, rotation errors are in degree, and translation errors are multiplied by 100. For Depth Estimation, we follow [39] to adopt the Mean Depth Absolute Error (MDAE) for comparing the rendered and the ground-truth depth values.

**Implementation Details.** We implement our framework based on previous work [1, 21]. For GC-NeRF, 1024 rays will be calculated in every batch. And we sample 128 coordinates along each ray with a single MLP network. We initialize camera extrinsic parameters by: the rotation matrix $\mathbf{R} \in \mathbb{R}^{3\times3}$ as $\mathbf{E} + \Delta_r$, the translation vector $\mathbf{t} \in \mathbb{R}^3$ as $\mathbf{0} + \Delta_t$. Moreover, we follow [1] to align test camera poses before the evaluation. As for baselines, we defer to the paper and source codes to conduct experiments with the same settings, which are under 2 input views and 20K iterations with about 4-5 hours on a single RTX 3090 GPU.

## 5.2 Comparing to SOTA with Unknown Pose NeRF

We evaluate GC-NeRF and the SOTA unknown pose methods with 2 input views in different metrics. Moreover, more details results are reported in the supplementary.

**Baselines.** We compare to four SOTA unknown pose baselines. **BARF** [17], an effective method with a simple strategy for training NeRF from imperfect camera poses. **SCNeRF** [13] is a self-calibration algorithm, which learns the scene and camera parameters jointly. **NoPeNeRF** [1] integrates depth additionally to establish relations between adjacent views for unposed NeRF. **SPARF** [39] jointly optimizes camera poses and scenes with three input views, which rely on noised rather than initializing poses with identity. For a fair comparison, we eliminate the GT depth in NoPeNeRF [1].

**Results of Novel View Synthesis.** As illustrated in Table 1, for the quantitative results, GC-NeRF outperforms all baselines in three datasets with different metrics by a large margin. Specifically, compared with SPARF in the Tanks and Temples dataset, GC-NeRF has increased by 2.05, 0.10, and 0.07 with respect to PSNR, SSIM, and LPIPS. This phenomenon demonstrates that the presented various consistency strategies can augment the ability of NeRF model from sparse and unposed images. For the qualitative analysis of the image rendering, GC-NeRF outperforms the SOTA unposed NeRFs significantly under two input views, which are shown in Fig. 2. This further illustrates the superiority of the geometric consistency framework to augment NeRF model and pose joint optimization.

**Results of Camera Pose Estimation.** The performances of pose estimation on three datasets are illustrated in Table 2. Compared with the SOTA approaches, our method achieves the best performance consistently. Specifically, in the Tanks and Temples

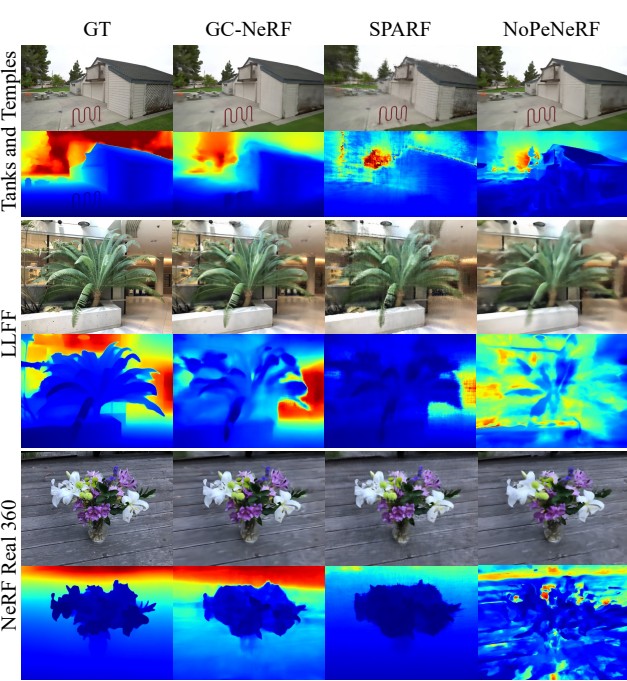

**Figure 2: The visualization of baselines and GC-NeRF with unknown pose images (2 views) about novel view rendering.**

scene, the accuracy of camera pose metrics for GC-NeRF is an order of magnitude higher than SPARF.

**Results of Depth Estimation.** We evaluate the rendered depth on Tanks and Temples, LLFF, and NeRF Real 360 datasets, respectively. As shown in the MDAE metric of Table 2, GC-NeRF performs superior accuracy for depth rendering, which precedes other baselines markedly. In addition, the qualitative visualization results on three datasets are exhibited in Fig. 2.

## 5.3 Comparing to SOTA with Known Pose NeRF

We report the performance of GC-NeRF and the SOTA known pose methods with 2 input views in novel view synthesis quality under fixed ground-truth camera parameters.

**Baselines.** We compare to four designed for tackling sparse image rendering methods, which are **DSNeRF** [6], **RegNeRF** [23], **SPARF** [39], along with the standard **NeRF** [21]. Especially, all baselines employ ground-truth camera poses.

**Results of Novel View Synthesis.** We exhibit the performance on three datasets in Table 3 and Fig. 3. Consequently, our approach GC-NeRF achieves the best performance compared to other baselines on all datasets and three rendering quality metrics. This phenomenon demonstrates that: Firstly, adopting image-level consistency covers the shortage of the traditional rendering losses, which enhances novel view synthesis performance. Secondly, the incorporation of region-level consistency significantly promotes NeRF's ability for depth estimation, thereby directly advancing the perception of complex scenes. Lastly, the AdaRFF positional encoding boosts high-frequency information within the images, maximizing the benefits of region consistency supervision. As a result, GC-NeRF achieves SOTA results even in the absence of GT poses.

**Table 1: The performance (2 views) of unknown pose methods about novel view synthesis in Tanks and Temples, LLFF, and NeRF Real 360 datasets.**

| Methods | Tanks and Temples | | | LLFF | | | NeRF Real 360 | | |
|---|---|---|---|---|---|---|---|---|---|
| | PSNR↑ | SSIM↑ | LPIPS↓ | PSNR↑ | SSIM↑ | LPIPS↓ | PSNR↑ | SSIM↑ | LPIPS↓ |
| BARF | 22.23 | 0.59 | 0.48 | 18.24 | 0.42 | 0.53 | 16.23 | 0.34 | 0.47 |
| SCNeRF | 19.39 | 0.56 | 0.43 | 18.60 | 0.41 | 0.44 | 15.05 | 0.31 | 0.52 |
| NoPeNeRF | 22.91 | 0.60 | 0.54 | 18.15 | 0.44 | 0.47 | 18.34 | 0.40 | 0.57 |
| SPARF | 22.45 | 0.59 | 0.47 | 19.75 | 0.46 | 0.54 | 20.06 | 0.44 | 0.43 |
| GC-NeRF | **24.50** | **0.69** | **0.40** | **23.00** | **0.64** | **0.29** | **21.07** | **0.52** | **0.39** |

**Table 2: The performance (2 views) of unknown pose methods about camera pose and depth estimation in Tanks and Temples, LLFF, and NeRF Real 360 datasets.**

| Methods | Tanks and Temples | | | LLFF | | | NeRF Real 360 | | |
|---|---|---|---|---|---|---|---|---|---|
| | $RPE_R↓$ | $RPE_t↓$ | MDAE↓ | $RPE_R↓$ | $RPE_t↓$ | MDAE↓ | $RPE_R↓$ | $RPE_t↓$ | MDAE↓ |
| BARF | 0.506 | 5.958 | 1.570 | 2.980 | 9.882 | 1.218 | 2.941 | 9.219 | 1.590 |
| SCNeRF | 0.526 | 1.822 | 0.893 | 3.412 | 3.244 | 0.907 | 2.140 | 2.840 | 0.890 |
| NoPeNeRF | 0.875 | 0.819 | 0.549 | 3.682 | 5.801 | 0.880 | 1.892 | 1.580 | 0.621 |
| SPARF | 0.406 | 0.603 | 0.365 | 0.611 | 2.217 | 0.514 | 0.387 | 0.811 | 0.213 |
| GC-NeRF | **0.115** | **0.215** | **0.358** | **0.412** | **0.334** | **0.233** | **0.233** | **0.476** | **0.171** |

**Table 3: The performance (2 views) of known pose methods about novel view synthesis in Tanks and Temples, LLFF, and NeRF Real 360 datasets.**

| Methods | Tanks and Temples | | | LLFF | | | NeRF Real 360 | | |
|---|---|---|---|---|---|---|---|---|---|
| | PSNR↑ | SSIM↑ | LPIPS↓ | PSNR↑ | SSIM↑ | LPIPS↓ | PSNR↑ | SSIM↑ | LPIPS↓ |
| NeRF | 15.78 | 0.42 | 0.65 | 16.29 | 0.31 | 0.58 | 15.77 | 0.22 | 0.70 |
| DSNeRF | 20.93 | 0.60 | 0.57 | 20.69 | 0.53 | 0.51 | 16.69 | 0.32 | 0.66 |
| RegNeRF | 20.20 | 0.65 | 0.58 | 21.39 | 0.64 | 0.58 | 19.22 | 0.50 | 0.61 |
| SPARF | 21.38 | 0.58 | 0.45 | 22.32 | 0.58 | 0.38 | 20.77 | 0.51 | 0.33 |
| GC-NeRF | **23.62** | **0.68** | **0.40** | **23.39** | **0.66** | **0.30** | **20.95** | **0.53** | **0.30** |

**Table 4: The ablation study for image and region-level consistency on LLFF dataset.**

| Variants | NVS | | | PE | |
|---|---|---|---|---|---|
| | PSNR↑ | SSIM↑ | LPIPS↓ | $RPE_R↓$ | $RPE_t↓$ |
| GC-NeRF | **23.00** | **0.64** | **0.29** | **0.412** | **0.334** |
| w/o $\mathcal{L}_{aps}$ | 19.42 | 0.50 | 0.39 | 1.032 | 0.875 |
| w/o $\mathcal{L}_{mps}$ | 18.28 | 0.43 | 0.41 | 1.831 | 1.389 |
| w/o $\mathcal{L}_{sss}$ | 16.96 | 0.36 | 0.46 | 2.574 | 1.490 |
| w/ ReProE | 17.64 | 0.39 | 0.43 | 2.390 | 1.411 |

**Table 5: The ablation study for positional encoding strategy on LLFF dataset.**

| Variants | NVS | | | PE | |
|---|---|---|---|---|---|
| | PSNR↑ | SSIM↑ | LPIPS↓ | $RPE_R↓$ | $RPE_t↓$ |
| w/ AdaRFF | **23.00** | **0.64** | **0.29** | **0.412** | **0.334** |
| w/ SPARF | 19.89 | 0.48 | 0.32 | 0.486 | 0.399 |
| w/ NoPeNeRF | 19.70 | 0.46 | 0.34 | 0.550 | 0.730 |
| w/ NeRF | 18.96 | 0.44 | 0.48 | 0.959 | 1.599 |
| w/o HiMF | 18.45 | 0.44 | 0.74 | 1.384 | 1.859 |

## 5.4 Method Analysis

In this section, we conduct a comprehensive analysis of the components and hyper-parameters that are essential to our approach. All methods are evaluated on the LLFF [33] dataset.

**Effect of Image and Region Level Consistency.** We ablate the key supervisions of our approach with novel view synthesis (NVS) and pose estimation (PE), here eliminating image- and region-level consistencies from GC-NeRF, respectively. We also replace the region-level supervisions with the re-projection error (w/ ReProE) to test the performance. The results are illustrated in Table 4. We observe that the first row achieves the best performance. Because

GC-NeRF establishes robustness correspondence among multi-view images, which render higher quality images and accurate camera poses simultaneously. Comparing the rest the rows with the standard NeRF [21] in Table 3, the formers achieve better results, which demonstrates that the image- and region-level geometric supervisions are effective for NeRF and pose joint optimization.

**Impact of Positional Encoding Strategy.** We examine the impact of the adaptive random Fourier feature (w/ AdaRFF) positional encoding on the image rendering and PE performances by removing or changing it to other alternatives. The results are listed in Table 5. Without AdaRFF, the alternative strategies are the coarse-to-fine

GT  GC-NeRF  SPARF  RegNeRF

Tanks and Temples

LLFF

NeRF Real 360

**Figure 3: The visualization of baselines and GC-NeRF with known pose images about novel view rendering. The inputs only contain 2 images.**

positional encoding strategy in SPARF [39] (w/ SPARF), the aperiodic trigonometric function in NoPeNeRF [1] (w/ NoPeNeRF), the standard Fourier transformation in NeRF [21] (w/ NeRF), and without any high-frequency mapping functions (w/o HiMF). Comparing the results of w/o AdaRFF, although three alternative mapping functions for the same backbone achieve better performance than w/o HiMF in representing texture and pose estimation. Nevertheless, GC-NeRF outperforms all above strategies because of aggregating richer high-frequency information adaptively according to different positions and viewing direction. Notably, compared the variants with Table 1 and Table 2, all variants achieve better performance. It verifies the superiority of the image- and region-level geometric consistency supervisions for NeRF and poses joint optimization.

**Amount of Training Data.** In Table 1, we evaluate our presented approach GC-NeRF for joint NeRF-pose optimization, when considering only 2 input views. For completeness, we here report results on LLFF data when 5 and 8 input views are available. The experimental settings are the same as 2-view settings, except for the number of training views. As illustrated in Table 6, the trend is similar for 2, 5, and 8 input images. Although SPARF [39] yields better performance as the images increase, it still struggles with camera poses. GC-NeRF outperforms SPARF consistently in three groups, and both the rendering quality and camera parameters are more satisfactory. Because the image- and region-level supervisions provide sufficient multi-view geometry consistencies, which facilitate the optimization of NeRF model and pose parameters simultaneously.

**Amount of Matching Points.** The matching points are the basis of GC-NeRF to construct region-level consistency. We evaluate the effectiveness of matching points with different quantities on the

**Table 6: The performance of GC-NeRF and SPARF with different numbers of training data on LLFF dataset.**

| Methods | NVS | | | PE | |
|---|---|---|---|---|---|
| | PSNR↑ | SSIM↑ | LPIPS↓ | RPE$_R$↓ | RPE$_t$↓ |
| Ours-2 | **23.00** | **0.64** | **0.29** | **0.412** | **0.334** |
| SPARF-2 | 19.75 | 0.46 | 0.54 | 0.611 | 2.217 |
| Ours-5 | **23.15** | **0.69** | **0.32** | **0.186** | **0.254** |
| SPARF-5 | 21.38 | 0.52 | 0.47 | 0.194 | 0.365 |
| Ours-8 | **23.82** | **0.73** | **0.28** | **0.088** | **0.242** |
| SPARF-8 | 21.81 | 0.54 | 0.46 | 0.193 | 0.320 |

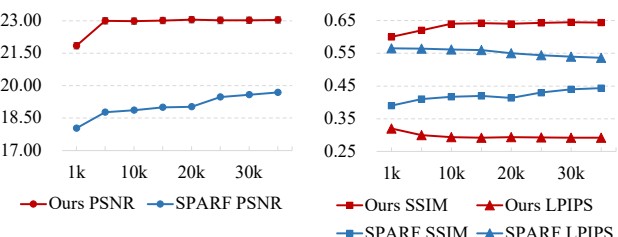

**Figure 4: The performance of GC-NeRF and SPARF with different numbers of matching points on LLFF dataset.**

LLFF scene. In Fig. 4, all metrics get better as the number of matching points gets bigger. The phenomenon depicts that both methods can improve the rendering quality as the matching gets denser. However, GC-NeRF will oscillate at a high level after reaching the threshold (i.e. 5k), because of calculating gradients for both pose and NeRF model. Yet SPARF need more than 35k to converge just for density field. Notably, the number of reliable matching points is extremely limited with viewpoint variations in the real-world scenario. It illustrates the availability and robustness of GC-NeRF.

## 6 CONCLUSION
Current methods for novel view synthesis struggle to render high-quality images and preserve high-frequency details, when trained from sparse and unposed images by the joint optimization of NeRF models and camera poses. This paper proposes learning geometry consistent neural radiance field from sparse and unposed views (GC-NeRF) for novel view synthesis. The presented GC-NeRF establishes the consistency relations from image- and region-level. The former adopts the epipolar geometry for generating pseudo-color labels to construct adjacent photometric supervision, which integrates intra- and inter-views photometric constraints to augment the gradients for both NeRF and camera poses. The latter employs matching points to capture the re-projected color labels for facilitating the joint optimization of the NeRF and poses. The proposed adaptive fusion random Fourier features positional encoding module maps the input coordinates to a high dimension space adaptively, which effectively fits the high frequency variation of the 3D scene. We evaluate GC-NeRF and several state-of-the-art methods in three challenging real-world scenarios. The experimental results verify the effectiveness and robustness of the GC-NeRF. We also plan to extend our strategy to other NeRF's hotspot applications (e.g. relighting with NeRF from sparse and unposed views).

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
