# OpenReview forum: "Learning Geometry Consistent Neural Radiance Fields from Sparse and Unposed Views"
_acmmm.org/ACMMM/2024/Conference — MM2024 Poster_

### Official Review · Reviewer_rZGS · 2024-04-30

**Rating:** 4
**Confidence:** 3

**Summary:**

This paper proposes GC-NeRF for novel view synthesis from sparse input images without known poses. It jointly optimizes NeRF models and camera poses with 1) image warping and 2) matching point warping. An adaptive positional encoding method is also introduced to improve high-frequency details. Experiments with 2 input images show that GC-NeRF achieves state-of-the-art results in multiple situations with unposed and sparse views.

**Strengths:**

The proposed GC-NeRF is a method to solve novel view synthesis under sparse and unposed views, which is a challenging task. By combining image-based and point-based warping techniques to add photometric and geometric constraints, this method achieves state-of-the-art performance in settings with sparse and unposed views.  The effects of each contribution are also well-validated according to experiments with image and geometry metrics, and the results seem promising.  This paper is easy to follow.

**Limitations:**

- **More views.**  The comparisons in the paper are mainly focused on settings with 2 views. Despite that Table 6 has shown the method's availability for settings with more unposed views and meanwhile can perform better than SPARF, it's unclear that: How to extend the proposed GC-NeRF from an image pair to multiple views? Is it an overall joint optimization, or pair-by-pair with multiple stages? And are there any additional hypotheses, like the images are sorted as a sequence (used in NoPeNeRF)? As the 2-view setting is rare in both previous works and real-world applications, it's essential to add an explanation about the details for multiple views.

- **Detailed experiment settings should be declared.** In section 4.2 Unknown Pose setting, it's unclear whether all baselines are with the same unknown pose as in the proposed method, or keep their original settings (e.g. using noisy poses for SPARF instead of unknowns).

- **Typos.**
  1) text format for the same symbols in eq (10) are inconsistent.
  2) the expressions in line 338 and eq (6) are suggested to be unified whether $[R \quad t]$ or $[R|t]$


- Although the combination of the contributions has a great effect, some similar insights can be found in previous works (e.g. monodepth and SPARF), and the learnable positional encoding is somehow similar to current learning-based encoders like instant-NGP. Nevertheless, this combination is also valuable for research.

Will consider increasing the rating if my concerns are well solved

**Suitability:**

2

---

### Official Review · Reviewer_8BYU · 2024-05-16

**Rating:** 3
**Confidence:** 2

**Summary:**

The paper presents a method called Learning Geometry Consistent Neural Radiance Field (GC-NeRF) for synthesizing views from sparse and unposed images. This method optimizes both a neural radiance field and camera poses simultaneously by establishing image-level and region-level geometric consistencies. It introduces an adaptive high-frequency mapping function to improve the 3D scene's geometric and textural details.

**Strengths:**

The numerical results looks good under the sparse setting.

**Limitations:**

1. Although mentioned in the limitation, I'm interested in a detailed comparison of computational costs and training times between the baseline methods and the proposed approach. This comparison would help understand the trade-offs in reconstruction quality against the computational cost required.
2. The authors stated that SPARF heavily depends on pose initialization. However, it's not entirely clear how the proposed method reduces this dependence with its new initialization strategy. I'm curious about how the proposed method performs under various initialization conditions, and how SPARF would perform if it used the same pose initialization as the proposed method.
3. The illustration appears somewhat cluttered, with numerous arrows pointing in different directions, suggesting the addition of many regularization terms to prevent overfitting rather than presenting a novel and elegant solution.

**Suitability:**

2

---

### Official Review · Reviewer_gxuH · 2024-05-21

**Rating:** 3
**Confidence:** 4

**Summary:**

This paper proposes GC-NeRF for jointly optimizing NeRF camera poses under sparse and unposed image sets. GC-NeRF use geometric consistencies between training views, i.e. warp one image to another camera pose according to depth. It also uses region-level consistency that enforce pixel-level consistency between keypoints at different viewpoints. Finally, GC-NeRF presents an adaptive high-frequency mapping as an alternative to original positional embedding in NeRF. Results show that the proposed method achieve superior performance on multiple challenging datasets.

image- and region-level consistencies

**Strengths:**

- The adaptive positional encoding strategy designed for augmenting high-frequency perception is novel and interesting
- The results look quite impressive for 2-view inputs

**Limitations:**

- To my knowledge, the geometric and matching projected supervision is not novel, similar ideas have already been used in many NeRF papers before [1][2] to deal with the sparse view setting, which this paper haven't cited.
- One question is that when the camera pose is not accurate at the begining of training, wouldn't enforcing these geometry constraints introduce a lot of errors?
- No comparisons against more recent methods like SparseNeRF [3]
- There are no enough comparisons on using other number of views, e.g. 3 view or 6 view with other baselines, making me unconfident about the efficacy of the method
- The proposed method seem to be a bit slow, needs 4 hours for each scene

[1] GeCoNeRF: Few-shot Neural Radiance Fields via Geometric Consistency
[2] StructNeRF: Neural Radiance Fields for Indoor Scenes with Structural Hints
[3] SparseNeRF: Distilling Depth Ranking for Few-shot Novel View Synthesis

**Suitability:**

2

---

### Official Review · Reviewer_ZRVN · 2024-05-24

**Rating:** 4
**Confidence:** 3

**Summary:**

This paper proposes GC-NeRF, a method for learning neural radiance fields from sparse and unposed views. It introduces image-level and region-level consistency losses to jointly optimize the NeRF model and camera poses. An adaptive positional encoding strategy is also presented to better capture high-frequency details. Through the introduction of multiple losses, GC-NeRF enables NeRF to learn and generate high-quality novel views solely from a sparse set of images.

**Strengths:**

- The paper is overall clear and well-motivated. The proposed method (GC-NeRF) employs multiple losses where the justification of using each of the losses is clearly written.
- GC-NeRF achieves state-of-the-art performance in multiple datasets, showing large performance improvements compared to prior works.
- The proposed adaptive positional encoding seems like a useful enhancement to capture high-frequency scene details.

**Limitations:**

Although the work shows promising performance in multiple datasets, I have a few concerns I would like the authors would resolve. With my questions resolved, I will change my rating to a higher rating.

- Although the introduction of the proposed adaptive position encoding is novel and convincing, I did not understand the big performance gap compared to using the frequency annealing strategy adapted in SPARF[1] and BARF[2]. The concern stems from the large performance gap shown in Table 3 of the paper, showing 3.1dB gap with the Sparf encoding. Specifically, I would like to see a theoretical comparison of the proposed AdaRFF vs Frequency Annealing.
- Similar to the previous concern, I would like to see the ablation of removing the AdaRFF in GC-NeRF’s architecture. Comparing Table 2 and Table 3, replacing AdaRFF seems to show similar results with SPARF. To understand the effectiveness of other components and AdaRFF, I would like to see additional experiments of using SPARF + AdaRFF.
- If my understanding is correct, $L_{aps}$ and $L_{mps}$ seem to be similar as the training converges. Specifically, when  $p_s$ has a similar depth with the $p_{s \rightarrow t}$, this indicates the high chance that the two pixels come from the surface of the scene. This implicitly confirms that the two pixels should correspond in the rendering as well. From this knowledge, is there any special reason for adopting both losses? It seems that using $L_{mps}$ is enough.

[1] Truong, Prune, et al. "Sparf: Neural radiance fields from sparse and noisy poses." Proceedings of the IEEE/CVF Conference on Computer Vision and Pattern Recognition. 2023.

[2] Lin, Chen-Hsuan, et al. "Barf: Bundle-adjusting neural radiance fields." Proceedings of the IEEE/CVF International Conference on Computer Vision. 2021.

**Suitability:**

2

---

### Meta-Review · Area_Chair_9Muw · 2024-07-03

**Recommendation:** Accept (Poster)
**Confidence:** 4

**Metareview:**

The submission received 4 final ratings, with only 1 being negative. Positive reviewers basically acknowledge the novelty. Reviewer 8BYU, who gave a negative rating, feels that this is a careful rebuttal and states that 'the reviewer's name in the rebuttal is wrong. Second, I don't understand what "They are comparable because RGB rendering is only 40K model parameters than depth."'

After carefully checking the rebuttal, the AC also found the mistake pointed out by Reviewer 8BYU. However, the AC can connect the rebuttal and the review even if the reviewer's names are incorrectly presented, and the concerns have been well addressed.

Considering that the contribution of this work outweighs such a mistake in the rebuttal, the AC decided to support this work on condition that the authors will follow the review to revise the paper in the camera ready version.